# Inadequate Annotation and Its Impact on Pelvic Tilt Measurement in Clinical Practice

**DOI:** 10.3390/jcm13051394

**Published:** 2024-02-28

**Authors:** Yuan Chai, Vincent Maes, A. Mounir Boudali, Brooke Rackel, William L. Walter

**Affiliations:** 1Sydney Musculoskeletal Health and The Kolling Institute, Northern Clinical School, Faculty of Medicine and Health and the Northern Sydney Local Health District, Sydney, NSW 2006, Australia; mounir.boudali@sydney.edu.au (A.M.B.); bill.walter@hipknee.com.au (W.L.W.); 2Department of Orthopaedics and Traumatic Surgery, Royal North Shore Hospital, St. Leonards, NSW 2065, Australia; drvincentmaes@gmail.com; 3Sydney Medical School, The University of Sydney, Sydney, NSW 2006, Australia; brac4608@uni.sydney.edu.au

**Keywords:** pelvic tilt, spinopelvic alignment, surgical templating, annotation accuracy, radiographic analysis, personalized surgery

## Abstract

Background: Accurate pre-surgical templating of the pelvic tilt (PT) angle is essential for hip and spine surgeries, yet the reliability of PT annotations is often compromised by human error, inherent subjectivity, and variations in radiographic quality. This study aims to identify challenges leading to inadequate annotations at a landmark dimension and evaluating their impact on PT. Methods: We retrospectively collected 115 consecutive sagittal radiographs for the measurement of PT based on two definitions: the anterior pelvic plane and a line connecting the femoral head’s centre to the sacral plate’s midpoint. Five annotators engaged in the measurement, followed by a secondary review to assess the adequacy of the annotations across all the annotators. Results: The outcomes indicated that over 60% images had at least one landmark considered inadequate by the majority of the reviewers, with poor image quality, outliers, and unrecognized anomalies being the primary causes. Such inadequacies led to discrepancies in the PT measurements, ranging from −2° to 2°. Conclusion: This study highlights that landmarks annotated from clear anatomical references were more reliable than those estimated. It also underscores the prevalence of suboptimal annotations in PT measurements, which extends beyond the scope of traditional statistical analysis and could result in significant deviations in individual cases, potentially impacting clinical outcomes.

## 1. Introduction

The practice of patient-specific templating prior to surgical interventions has become a mainstay for many surgeons [1]. This process often necessitates an examination of radiographic landmark annotations [2,3]. Pelvic tilt (PT) is a routinely measured radiographic parameter in hip and spine surgeries [4,5] for assessing spinopelvic alignment and implant navigation [6,7].

In current clinical practice, a single annotator manually evaluates these anatomical landmarks and calculates the corresponding parameters in a high-pressure clinical environment [8]. The susceptibility to human error in these measurements is a well-acknowledged concern, and the clinical implications of these errors have not been fully explored [9]. The inherent subjectivity of these assessments, coupled with the ambiguity of landmark definitions, variations in patient anatomy, and inconsistencies in radiographic quality, contributes to this challenge [10]. Existing studies on the reliability of radiographic annotations focus on a parameter-level analysis, comparing the accuracy of lengths or angles of paired landmarks through statistical methods like mean absolute error, correlation, and reproducibility analyses [11,12,13,14]. Researchers establish thresholds to determine the reliability of a measurement dataset [15,16]. However, this approach may overlook instances where inadequate landmark annotation does not markedly affect the overall parameter, potentially obscuring landmark-specific inadequacies which could compromise patient care. To our knowledge, analyses that delve into the accuracy of radiographic landmark annotation at a point-wise level are absent from the literature.

In the domain of spinopelvic radiographic analysis, the literature reveals mixed findings on the measurement reliability of various spinal sagittal parameters among experienced surgeons. Some studies have highlighted unfavourable reliability due to the poor visualization of anatomical landmarks [10,17], while others assert the high reliability of these radiographic measurements [16]. Despite this academic debate, clear guidelines for enhancing the accuracy of landmark annotations in clinical settings remain elusive. Research has identified factors such as pathological changes and obesity that obscure certain landmarks [17,18], and different anatomical regions within a single image may exhibit varied error patterns [18,19]. Yet, the impact of suboptimal annotations at a landmark-specific level has not been thoroughly investigated [20]. To address these challenges, some researchers have suggested alternative parameters and the use of image augmentation techniques to improve measurement precision [19,21]. However, the body of work specifically focused on evaluating the accuracy of PT landmark annotations is notably sparse. In the rapidly advancing domain of artificial intelligence (AI), particularly within deep learning frameworks, popular semantic segmentation algorithms often yield landmark predictions as heatmaps, reflecting the model’s confidence in its assessments at a point-wise dimension [22,23]. Despite the widespread adoption of such models, there remains a notable gap in the analysis of point-wise accuracy pertaining to human-generated landmark annotations. This omission leads to an inadvertent incorporation of label noise into AI models, resulting in a fundamental lack of understanding of such noise. The current literature cites “precise” PT measurements by comparing AI-derived results with “gold standard” datasets produced through manual image annotations [24]. This approach, however, neglects the inherent uncertainties and potential inadequacies in manual annotation. Such oversights can lead to the misattribution of deviations from the gold standard to AI inaccuracies, disregarding the possibility that the “gold standard” itself may harbor errors stemming from human annotation [25]. Considering the discussions presented in the current literature, it is evident that there is a pressing need to understand the inadequacies inherent in human landmark annotation at a point-wise level.

This study aims to bridge the gap by conducting an accuracy analysis of PT landmark annotations on a landmark-wise basis. To accomplish this goal, we recruited multiple annotators to measure PT parameters in simulated clinical settings and then performed secondary reviews to collectively evaluate the annotations from different annotators. This approach aims to identify and rectify instances of inadequate annotation and ultimately provide insights into the accuracy of PT landmark annotations, elucidate the specific factors contributing to annotation inadequacies, and determine their impact on PT parameters.

## 2. Materials and Methods

### 2.1. Study Design

This retrospective study collected 126 consecutive sagittal radiographs from an academic surgical clinic (EOS Imaging, Paris, France [26]). Patients were enrolled between November 2020 and July 2021. Eleven radiographs were subsequently excluded due to various reasons: three for not meeting the clinical image quality standards, seven due to the presence of bilateral implants obscuring landmarks, and one owing to a disease (developmental dysplasia) that could influence the measurements. Thus, a total of 115 lateral pelvic radiographs from 93 consecutive patients (62 males and 31 females, aged 64.6 ± 11.4 years) were included. The de-identified patient data were collected from a research database that was ethically approved by the St. Vincent’s Hospital Human Research Ethics Committee (2019/ETH09656) in Sydney, with all the participants providing informed consent for the use of their anonymized data for research purposes.

### 2.2. Landmark Annotations

Five annotators engaged in the landmark annotation process, including two orthopaedic engineers, two orthopaedic fellows, and one surgeon. Each annotator possessed a minimum of 100 pre-surgical templating experiences, ensuring their competency in measuring corresponding parameters. All the annotations were conducted using a customized code in the Image Processing Toolbox in MATLAB (2022b MathWorks, Natick, MA, USA). Two pelvic tilt definitions were used (Figure 1) [7]. The anatomical definition (PT_a_) comprises the gravity line and the line connecting the centre of the anterosuperior iliac spines (ASISs) and the pubic tubercles. The mechanical definition (PT_m_) comprises the gravity line and the line connecting the centre of the femoral heads and the midpoint of the sacral plate [7].

In a manner similar to standard clinical procedures which utilizes digital radiographs from the picture-archiving and communication system (PACS), annotators enlarged each image on the computer screen to a level where the interested anatomy region was sufficiently discernible, thereby enabling them to place all landmark points with confidence. The PT_a_ annotations were made by marking one point for the centre of the anterosuperior iliac spines (ASISs) and another for the pubic symphysis. The PT_m_ annotations were conducted via two methods: (1) the calculation method—three points for each femoral head contour (six points total) and two points for the anterior and posterior ends of the sacral endplate; and (2) the estimation method—one point for the centre of two femoral heads and one point for the midpoint of the sacral endplate. These coordinates were subsequently utilized to compute the PT parameters. In instances where it was feasible, the PT_m_ obtained semi-automatically by radiographic technicians via the sterEOS^®^ software (EOS Imaging, Paris, France) were collected for comparison with our manually annotated outcomes [26].

Each image received five separate annotations from independent annotators, each depicted in different colours (representative of each annotator) and shapes (circle for the calculation method and triangle for the estimation method, Figure 2). The fully annotated images were then subject to an error analysis.

### 2.3. Error Analysis

After a four-week interval from the completion of the annotation process, three assessors (one surgeon, one orthopaedic fellow, and one orthopaedic engineer) conducted an evaluation of all the images containing the five distinct annotations. This assessment was performed in a manner that was blinded to the identity of the annotators, as exemplified in Figure 2. By comparing each other’s annotations, the assessors were able to examine the distribution of the landmarks and gain a collective insight into the underlying causes of inadequate annotations. They rated each landmark based on the following: (1) all five annotations meet the clinical standard for a landmark (all satisfactory); or (2) specific annotations failed to meet the clinical standard due to a low image quality at the landmark region, the presence of anatomical anomalies, annotations away from the intended target (outliers) while the target is identifiable, or due to other factors that could not be categorized within the aforementioned conditions. The assessments were recorded on a custom survey form via REDCap (Vanderbilt University, Nashville, TN, USA).

Upon the initial data collection and review, we adopted a “majority rules” approach for data categorization, where opinions from two or more assessors were regarded as the ground truth. The inadequacies in landmark annotation were analysed in the following manner:(1)Landmark-wise inadequacy: instances where two or more assessors identified at least one out of five annotations of a landmark as inadequate.(2)Reason-wise inadequacy: cases where two or more assessors highlighted the same reason for a landmark’s inadequacy.(3)Parameter-wise impact: if two or more assessors identified the same annotation (both colour and shape) as inadequate, the corresponding annotation and its PT parameter were subsequently excluded from the “adequate” dataset group. Comparative statistics between the “full” dataset group and the “adequate” dataset group were then conducted.

Consider Figure 2 as an illustrative case: the annotations for the centre of the femoral heads and the midpoint of the sacral plate did not exhibit any discernible outliers. Although not perfectly overlapping and with a few annotations potentially neglecting the contour of the second femoral head on the right-hand side due to the predominance of overlapping contours, these annotations were clinically deemed adequate. In contrast, the annotations for the centre of the ASISs revealed notable discrepancies. The black dot clearly overlooked the ASIS on the right side, and the purple dot’s placement was too imprecise; thus, both were classified as inadequate outliers for this study. These inadequate annotations, identifiable by most surgeons and radiologists, had minimal impact on the PT_a_ parameter, which explains why traditional statistical methods failed to detect them. The blue dot, positioned slightly above the clustered red and green dots, was considered adequate due to its location within the centre of the two anterosuperior curves of the iliac spines—a region which allows for some subjective interpretation. Regarding the pubic symphysis region, only the red dot risked being an outlier, located at the anterosuperior edge of the pubic tubercles, whereas the other four dots were aligned at the anterior end of the curvature. Definitions for this landmark may vary across studies [27], and these variations are generally not seen as affecting clinical judgment. Consequently, all annotations at the pubic symphysis region were judged to be adequate.

### 2.4. Statistical Analysis

The PT parameters for both the “full” dataset group and the “adequate” dataset group were evaluated. The case-wise average parameters were compared using a correlation analysis, the mean absolute difference, the maximum absolute difference, 95% confidence intervals (CIs), and paired t-tests. Pearson’s correlation coefficient was interpreted as “poor” for *r* < 0.3, “fair” for 0.3 < *r* < 0.5, “moderate” for 0.5 < *r* < 0.6, “moderate strong” for 0.6 < *r* < 0.8, and “very strong” for *r* > 0.8 [28]. The reliability of the PT_m_ measurements was validated using the intraclass correlation coefficient (ICC) by comparing the measurements with the values reported by the radiographic technician [29]. An ICC above 0.9 was interpreted as representing an excellent agreement [29]. All statistical analyses were conducted using SPSS (IBM, Tulsa, OK, USA).

## 3. Results

Our landmark-wise inadequacy analysis indicated that, in 61.74% of cases, at least one landmark was not deemed “all adequate” (Table 1). This means that, for at least one landmark per image, the majority of assessors identified it as inadequate for various reasons. Of these, both the ASIS and the femoral head centre (determined using the estimation method) contained inadequate annotations in over 30% of cases, as evaluated by two or more assessors. Annotations based on clearly identified anatomical contours and calculated landmark positions demonstrated a higher accuracy, with lower instances of inadequacy, compared to those relying on estimations of landmark locations. Specifically, the femoral head centre showed a 7.83% inadequacy rate for the identified anatomical positions versus 30.43% for the estimated positions, and the midpoint of the sacral plate had an 11.30% inadequacy rate compared to the 13.04% rate for the estimated positions.

Our in-depth reason-wise analysis revealed that landmarks in different anatomical regions displayed distinct error tendencies (Table 2). Annotations of the ASIS frequently proved to be inadequate due to a compromised image quality and an outlying location. In cases estimating the femoral head centre, a higher incidence of outlier annotations was observable when examined with annotations from multiple assessors. Additionally, anomalies associated with the sacral plate were occasionally overlooked (in 5–7% of cases). The “Other” reason associated with the two cases was an excessive “axial rotation” that made the radiograph not sagittal enough for assessment. Further explanation of these assessments is provided in the Section 4.

A subsequent analysis revealed that inadequate annotations typically resulted in measurement discrepancies ranging from −2° to 2° (95% CIs, Table 3). The mean absolute difference observed between the “full” dataset group and the “adequate” dataset group was minimal, ranging only from 0.35° to 0.52°. Despite this, all the correlation coefficients exceeded 0.9, and the paired *t*-test revealed no statistically significant differences (all *p*-values above 0.05). Notably, the maximum difference observed in our cohort reached a value as high as 13.47°.

Among the 115 images, the radiographic technician reported PT_m_ measurements for 113 images using the sterEOS^®^ software. The reliability analysis of the measurements indicated an excellent agreement between our manual methods and the software’s automated measurements, with ICC values ranging from 0.91 to 0.94.

## 4. Discussion

Our study sheds light on the potential inadequacies in the current practice of radiographic landmark annotation, specifically related to the parameters of pelvic tilt in hip and spine surgeries [2]. A significant finding was that, in a dataset in which the measurement reliability was deemed as excellent as in traditional settings, more than 60% of cases analysed still contained at least one landmark annotation rated as inadequate by two or more assessors. This result underscores the necessity to review and potentially revise current annotation practices [21,30]. Intriguingly, the anatomical landmarks of the ASIS and the femoral head centre (when determined by estimation) were frequently deemed inadequately annotated in over 30% of cases, which underlines the prevalence of substandard radiographic landmark annotations in these regions [20,31]. Notably, the use of estimation in determining landmark locations seemed to be a significant contributor to this inadequacy. In contrast, more accurate anatomical measurements were obtained when more anatomical features were annotated by identifying anatomical contours and calculating landmark locations.

Our detailed reason-wise analysis helped unravel the reasons behind these inadequacies. When estimating the centre of femoral heads and annotating the centre of ASISs, annotations were frequently deemed outliers because the contralateral side of the bone was overlooked [31,32]. This was often due to the assumption that sagittal radiographs were strictly “sagittal”, leading to the misidentification of one side of the bone as overlapped underneath the other [33]. The assessors and annotators observed that poor image quality and outlier annotations in the ASIS region were primarily due to patients’ high BMI, which obscured the belly region and complicated the annotation process [18,31]. The literature has also highlighted that pathologic changes associated with femoral heads likely contribute to the low accuracy of femoral head centre identification by estimation [17]. Although our study did not demonstrate this impact, future studies could explore whether specific pathologies have a higher propensity for inaccuracies in femoral head centre identification. The underestimation of anomalies associated with the sacral slope further underscores the need for thorough anatomical knowledge in the annotation process [34]. Specifically, a lumbosacral transition and variations in the fusion of L5 and S1 are prevalent in the general population; thus sacral slope (SS) was often inaccurately measured by surgeons [17]. The literature estimates the prevalence of lumbosacral transitional vertebrae to be between 4.0 and 35.9%, with a mean of 12.3%, a value which is close to our outcomes [19]. Given the widespread nature of anatomical variations in this region, it is imperative for annotators to incorporate it into their deliberations when discerning sacral landmarks. Moreover, factors like the presence of hardware, anatomical deformities, and calibration issues with imaging equipment can adversely affect measurements [20]. However, such cases were excluded from our cohort, in line with our clinical protocols.

The traditional statistical analysis of the data reveals that inadequate annotations typically led to PT discrepancies ranging from −2° to 2° in most instances. The analysis showed a minor mean difference (0.35° to 0.52°) and a very strong correlation (above 0.9), with no significant statistical differences observed (all *p*-values > 0.05). These findings indicate that the measurement dataset adheres to clinical standards, even when including annotations deemed to be inadequate in a secondary review. Consequently, inadequate annotations appear to have exerted a minimal impact on the overall parameter analysis of our dataset. This suggests that suboptimal annotation practices may have limited effects on overall clinical decisions [30,35]. Although this result is supported by other studies [31,34,36], extreme errors could potentially influence clinical outcomes, especially in surgeries requiring high-precision measurements [2,32]. Therefore, the reasons behind these inadequacies should be flagged to improve both the quality of education and the data, particularly as these errors could inadvertently influence the emerging domain of machine learning algorithm training. Notably, although Dimar et al. identified inconsistencies in the manual measurements of sagittal parameters between surgeons, they found that a computer-aided programs yielded consistent, reliable measurements with a high-degree of correlation. When comparing the surgeons’ measures with the computer-aided measures, they found poor correlation in most measurements [10,37]. Our findings further highlight the irregularities in manual measures and strengthen the rationale for the implementation of machine-based measurements in the future.

The excellent agreement observed in the comparison of PT_m_ measurements across 113 images, between our manual annotations and those generated by the sterEOS^®^ software, verifies the effectiveness of our manual annotation techniques relative to automated software measurements. The sterEOS^®^ software, equipped with semi-automated statistical shape recognition, a divergence correction algorithm, and 3D parameter calibration with options for manual adjustments by radiographic technicians, demonstrated no significant statistical differences when compared to our method of radiographic annotation utilizing the MATLAB Image Processing Toolbox. This finding highlights the comparability of sophisticated automated systems with meticulous manual annotations, emphasizing the precision and reliability of human expertise in the context of radiographic measurements and analysis.

Despite these findings, several limitations of this study must be noted. First, our study did not consider the differing clinical experiences among annotators and assessors or their personal preferences during radiographic analysis. These differences could have affected the process of radiographic annotation and might be worth investigating in future studies. Second, the design of our assessment questionnaire remained subjective, and the decision to follow a “majority rule” over an “authority rule” method after consultation with a senior surgeon might not be applicable to other studies. Third, this study did not further categorize the reasons behind inadequate annotations, such as bad image quality due to patient movement, obesity, or low-dose X-rays, and outlying annotations due to wrongly identifying the contralateral bone, a compromised image quality, or carelessness. While distinguishing these subcategories could offer additional scientific insights, the subjective nature of image assessment and the potential for overlapping causes prevent the reliable isolation of each factor. Consequently, this study did not attempt to quantitatively differentiate the inadequate annotations based on these potential underlying reasons. Fourth, the radiographs in this study were produced using a fan-beam radiation source. Despite this newer radiographic technology offering lower radiation doses and having been demonstrated to be equivalent to traditional radiography [38,39]—hence its routine use in our clinical practice—the replication of this study using traditional cone-beam radiography would be beneficial. Such replication could help confirm the applicability and generalizability of our findings to conventional radiographic techniques. These factors may warrant further consideration in future research.

## 5. Conclusions

This study highlights the high incidence of suboptimal radiographic annotations in measuring pelvic tilt parameters, emphasizing that such inadequacies persist even in datasets demonstrating excellent measurement reliability with minimal clinical impact on the larger population. It is essential to acknowledge that, in individual cases, the maximum measurement discrepancies can reach up to 13°, a deviation substantial enough to potentially alter clinical outcomes [40]. Consequently, the prevalence of inadequate annotations cannot be ignored, and our results highlight important avenues for improving the quality of clinical practices related to this issue.

To enhance radiographic annotation practices in PT measurements, ensuring high-quality radiographs is fundamental. Improving these practices involves several targeted strategies:(1)Rigorous identification of anatomical landmarks: Practitioners should endeavour to precisely identify anatomical contours and calculate landmark locations with accuracy. This involves not approximating landmark locations based on visual estimates but utilizing well-defined anatomical markers within small, precise, and clear regions for calculation. This meticulous approach ensures that landmark identification is based on tangible anatomical features to minimise subjective interpretation.(2)Attention to bilateral anatomy: Special attention should be given to examining both sides of the anatomy, particularly the contralateral bones. This practice aids in recognizing and compensating for overlooked critical features that inform the precise location of landmarks. It prevents the assumption that the contralateral anatomy is fully overlapped.(3)Enhanced recognition of anatomical anomalies: Developing comprehensive knowledge for identifying anomalies that could alter the appearance of anatomical structures is crucial. Such anomalies may affect the precision of landmark identification and, consequently, the accuracy of PT measurements.

By adopting these practices, clinicians and radiographers can enhance the accuracy of measuring PT parameters, thereby enhancing clinical decision making in spinopelvic surgery. Additionally, our analysis contributes to the foundation of orthopaedic education by identifying inadequacies in annotations that go beyond what traditional statistics can reveal [1]. Furthermore, these improvements are not just limited to clinical applications, they also play a vital role in advancing the field of medical imaging by providing high-quality data for training automated deep learning models in landmark localization. This synergistic improvement across both manual and automated processes underscores the importance of precise and careful radiographic annotation in the broader context of medical imaging and orthopaedic care.

Future research should consider investigating the influence of annotator experience on the precision of landmark annotations, developing a more objective protocol for establishing gold-standard annotations, and delving deeper into the underlying causes of inadequate annotations, including the determinants of outliers and poor image quality, as well as their subsequent effects on radiographic parameters’ accuracy.

## Figures and Tables

**Figure 1 jcm-13-01394-f001:**
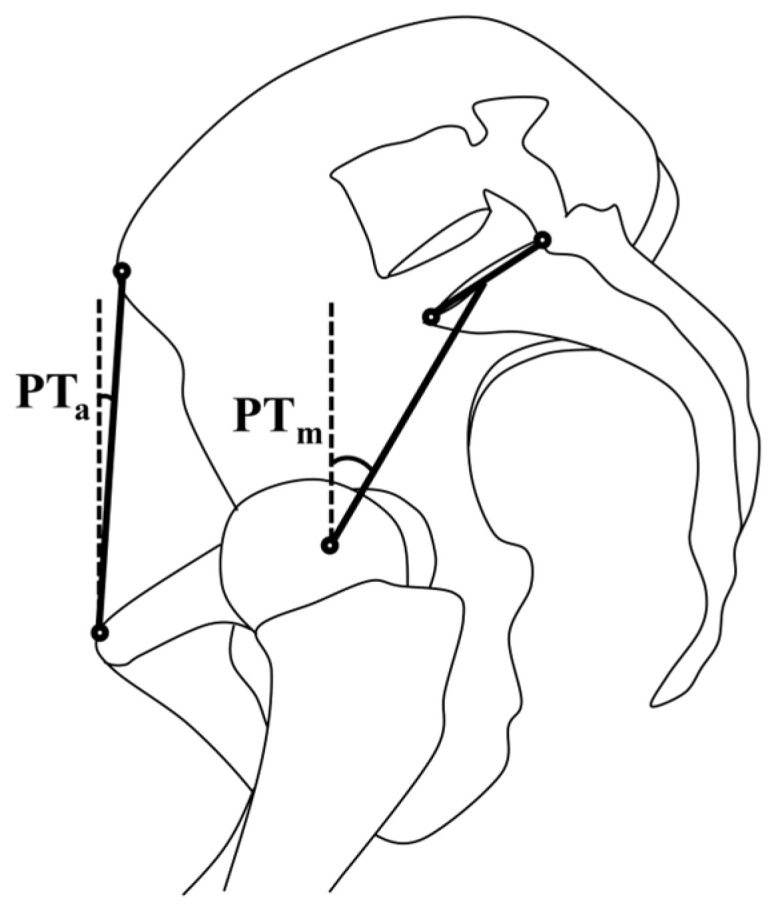
The anatomical definition (PTa) and mechanical definition (PTm) of pelvic tilt angles on sagittal pelvic radiographs.

**Figure 2 jcm-13-01394-f002:**
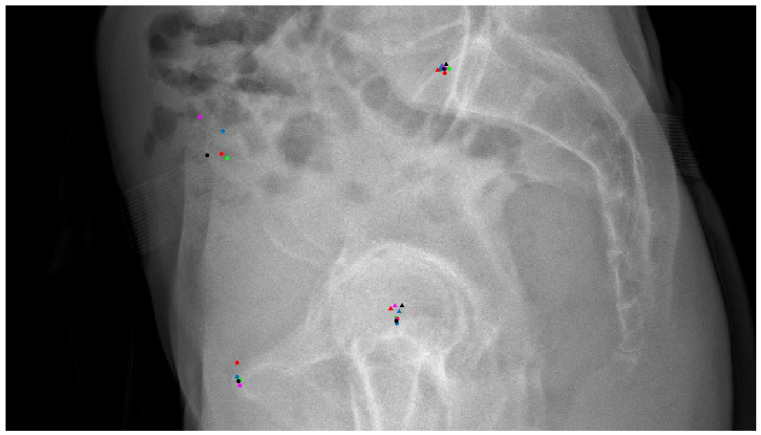
The radiographs with annotations from five annotators for a secondary assessment. Different colors indicate distinct annotators. Round dots represent annotations placed by directly clicking on the relevant location, while triangles represent annotations derived from calculating locations based on annotated bone contours.

**Table 1 jcm-13-01394-t001:** Cases containing inadequate annotations.

	ASIS	Pubic Tubercle	Femoral Head Centre (cal)	Femoral Head Centre (est)	Midpoint of Sacral Slope (cal)	Midpoint of sSacral Slope (est)	At Least One Landmark Was Inadequate
Cases contain inadequate annotation	42	11	9	35	13	15	71
Percentages	36.52%	9.57%	7.83%	30.43%	11.30%	13.04%	61.74%

cal = measured from bone contour annotations and calculating the location; est = measured from estimating the landmark location.

**Table 2 jcm-13-01394-t002:** Cases of inadequate annotations categorized by underlying reasons.

Landmark	Reason	Cases	Percentages
ASIS	Bad quality	17	14.78%
	Anomaly	0	0%
	Outlier	40	34.78%
	Other	2	1.74%
Pubic tubercle	Bad quality	3	2.61%
	Anomaly	0	0%
	Outlier	9	7.83%
	Other	0	0%
Femoral head centre (cal)	Bad quality	0	0%
	Anomaly	0	0%
	Outlier	9	7.83%
	Other	0	0%
Femoral head centre (est)	Bad quality	2	1.74%
	Anomaly	0	0%
	Outlier	35	30.43%
	Other	0	0%
Midpoint of sacral slope (cal)	Bad quality	0	0%
	Anomaly	6	5.22%
	Outlier	7	6.09%
	Other	1	0.87%
Midpoint of sacral slope (est)	Bad quality	0	0%
	Anomaly	8	6.96%
	Outlier	8	6.96%
	Other	0	0%

cal = measured from bone contour annotations and calculating the location; est = measured from estimating the landmark location.

**Table 3 jcm-13-01394-t003:** Parameter-wise comparison between the full dataset and the “adequate” dataset.

	PT_m__cal	PT_m__est	PT_a_
Correlation coefficient	0.98	0.99	0.99
Mean absolute difference	0.39°	0.35°	0.52°
Maximum difference	10.64°	13.47°	5.55°
−95% confidence interval	−2.16°	−1.93°	−1.92°
95% confidence interval	2.22°	2.47°	1.93°
*p*-value of the paired *t*-test	0.86	0.25	0.96

cal = measured from bone contour annotations and calculating the location; est = measured from estimating the landmark location.

## Data Availability

Measurement data are freely available under the CC0 license. De-identified imaging data and parameter data are deposited at Figshare for research non-identifiable purposes only (https://doi.org/10.6084/m9.figshare.23938398 accessed on August 2023). The transfer, storage, and use of radiographic data must follow our ethics approval (2019/ETH09656).

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
