# Peer review of "Inadequate Annotation and Its Impact on Pelvic Tilt Measurement in Clinical Practice"

_jcm, 2024, doi:10.3390/jcm13051394_

Round 1

Reviewer 1 Report

Comments and Suggestions for Authors

Dear authors,

I am pleased to review the submitted paper entitled "Inadequate Annotation and Its Impact on Pelvic Tilt Measure-2 ment in Clinical Practice".

The present paper focus on multiple annotators in  the measurement of PT parameters.

In my opinion the content is original, current, objective and persuasive. But there are some questions need to be address:

1. Study Design: “Patients were enrolled between November 2020 and July 2021 61 (EOS Imaging, France [16]).”Here need

to be specified that the position when patients take the exam, the position of the trunk,  upper body, lower body and balance situation should also be clearified. 

2. Study Design: "This retrospective study collected 126 consecutive sagittal pelvic radiographs from 60 an academic surgical clinic."  General information of the patients should be presented, including average age, gender need to be added. 

3.Landmark Annotations:"Five annotators engaged in the landmark annotation process, consisting of two or-70 thopaedic engineers, two orthopaedic fellows, and one surgeon."  Different annotators with different position(supine, two feet stand, one foot stand) should be compared.

4.Figure: A picture illustrate position when patient take examination should be added.  

Comments on the Quality of English Language

Minor editing of English language required.

Author Response

Thank you for dedicating your time and expertise to review our manuscript and for providing valuable suggestions. Please find attached the document named "Responses to Reviewer #1.docx" for detailed responses to each of your comments.

With sincere thanks,

Reviewer 2 Report

Comments and Suggestions for Authors

Thank you for allowing me the opportunity to review this manuscript that examines the reliability of spinal sagittal parameter measurements. This manuscript may be of potential interest considering that this is an aspect of spine imaging that has not been sufficiently studied, but I feel that the information provided in the present form is not of sufficient quality to merit publication in JCM. I would like to ask the authors to provide further analysis in order to improve the quality of the study.

The following are specific points that this reviewer feels needs to be improved/revised in this manuscript.

1. What is the novelty of this study? I agree that this topic has not been sufficiently studied, but there have been a number of published studies that have examined this topic. In the introduction, the authors should introduce the readers to more past studies and describe the novelty of their own examinations. Currently, I cannot discern any clear novel points of this manuscript.

2. Concerning methodology, are the annotators and assessors different people? Do all of these people practice in the clinical field of spine pathology who would regularly conduct these measurements of the spine? This should be made clear, because there are some several annotation points in Figure 2 that make me suspect that there may be some annotators who are not well versed in measuring spinal sagittal parameters.

3. I am not aware of the utility and limitations of the EOS Imaging system, but my impression is that it is a superior system to acquire whole spine radiographs compared to standard long film radiography. Why were there three images that did not meet clinical imaging quality standards? Furthermore, I am under the impression that the EOS software can automatically perform annotations and calculations of spinal sagittal parameters. Is this is correct, I ask the authors to consider comparing the results with the manual measurements of the annotators. 

4. In the analysis of the annotations, the specific reasons for a annotation to be categorized as an outlier are not well described. Is it just because one or two of the annotations were placed outside the general area of the majority of annotations? Please explain in more depth. 

Along the same lines, the following sentences (Line 131-134) should also be further explained:

Annotations of the ASIS frequently proved inadequate due to compromised image quality (Why? Were the anterosuperior iliac spines of the pelvis not discernable?) and outlying location (What does this mean? The authors should explain why they think it was placed in an outlying location). In cases estimating the femoral head center, a higher incidence of outlier annotations was observable when examined with annotations from multiple assessors (What does this mean?).

The authors do allude to some of the reasons in Lines 164-172, but the authors do not provide data on this point. What was the frequency of annotators not recognizing that the images were not strictly sagittal? What was the frequency that the annotators/assessors deemed a point to be difficult due to high BMI, and which points were specifically found to be difficult in overweight subjects?

5. I urge the authors to consider examining some of the points raised as limitations in Lines 200-210, because I feel that it could be novel contributions to the current literature. Without further examinations of these points, I am not convinced that this report is of sufficient quality to be published in JCM.

6. References 10 and 11 are redundant. Please amend. 

Comments on the Quality of English Language

There are no major issues with English.

Author Response

Thank you for dedicating your time and expertise to review our manuscript and for providing valuable suggestions. Please find attached the document named "Responses to Reviewer #2.docx" for detailed responses to each of your comments.

With sincere thanks,

Round 2

Reviewer 1 Report

Comments and Suggestions for Authors

All of my questions have been appropriately answered and modified, my conclusion is accept.

Author Response

Thank you for your review and acceptance. We greatly appreciate your feedback and support.

Reviewer 2 Report

Comments and Suggestions for Authors

The authors have made the effort to address all issues raised by this reviewer, and I believe that this manuscript has benefitted from this round of revision. 

While I still believe that this manuscript would be further improved by including an analysis of one or more points raised as limitations, I understand the authors desire to concentrate on the presented data.

I would like to ask the authors to further elaborate on the comparison with the manual annotations and the sterEOS software, because many surgeons do not have access to this system. While the authors state that there was excellent agreement between manual annotations and the sterEOS software, I believe it would benefit the readers if the authors could provide information on any points that demonstrated larger deviations, i.e., points that are more prone to disagreement, so that the readers can make the effort to examine the images carefully when annotating those points. 

Author Response

Thank you for your constructive feedback and guidance, which have significantly contributed to the improvement of our manuscript.

We recognize the concern regarding the accessibility of the sterEOS software, acknowledging that it is not universally available. Consequently, we designed our data collection methodology to be applicable across any PACS or annotation software, ensuring broader applicability and replicability. It's important to note that the sterEOS radiographic report is limited to a single PTm value without providing specific coordinates for each landmark. This limitation precludes a direct point-wise accuracy comparison between our manual annotations and those generated by the sterEOS software. Therefore, our study can only establish annotation guidelines based on our collected dataset.

We acknowledge the historical focus on accuracy and agreement analyses of paired landmarks within radiographic annotation research. By highlighting the need for a detailed examination of annotation accuracy at the individual landmark level, our study aims to encourage further research in this area, emphasizing the importance of precision in landmark annotation for enhancing the quality of clinical and research outcomes.